# Beo v1.0: Numerical model of heat flow and low-temperature thermochronology in hydrothermal systems.

Elco Luijendijk[1]

[1]Geoscience Centre, University of Göttingen, Goldschmidtstrasse 3, 37077, Göttingen, Germany

**Correspondence:** Elco Luijendijk (elco.luijendijk@geo.uni-goettingen.de)

**Abstract.** Low-temperature thermochronology can provide records of the thermal history of the upper crust and can be a valuable tool to quantify the history of hydrothermal systems. However, existing model codes of heat flow around hydrothermal systems do not include low-temperature thermochronometer age predictions. Here I present a new model code that simulates thermal history around hydrothermal systems on geological timescales. The modelled thermal histories are used to calculate apatite (U-Th)/He (AHe) ages, which is a thermochronometer that is sensitive to temperatures up to 70 °C. The modelled AHe ages can be compared to measured values in surface outcrops or borehole samples to quantify the history of hydrothermal activity. Heat flux at the land surface is based on equations of latent and sensible heat flux, which allows more realistic land surface and spring temperatures than models that use simplified boundary conditions. Instead of simulating fully coupled fluid and heat flow, the code only simulates advective and conductive heat flow, with the rate of advective fluid flux specified by the user. This relatively simple setup is computationally efficient and allows running larger numbers of models to quantify model sensitivity and uncertainty. Example case studies demonstrate the sensitivity of hot spring temperatures to the depth, width and angle of permeable fault zones, and the effect of hydrothermal activity on AHe ages in surface outcrops and at depth.

## 1 Introduction

The interpretation of thermochronological data relies on assumptions or models of the Earth's temperature field (Dempster and Persano, 2006). Thermal model codes that are used to interpret thermochronological data can take into account many processes that influence the temperatures in the Earth's crust on geological timescales, such as heat conduction, advection caused by the movement of faults blocks or changes in topography (Braun et al., 2012). However, these models do not include the thermal effects of groundwater flow, despite indications that groundwater flow often influences temperatures in the upper crust and thermochronological data sets (Ehlers, 2005; Ferguson and Grasby, 2011). A model study by Whipp and Ehlers (2007) forms an exception and demonstrated that diffuse topography-driven groundwater flow can strongly affect low-temperature thermochronometers in mountain belts.

While the thermal effects of fluid flow can complicate the interpretation of thermochronolgical datasets, low-temperature thermochronometers can also be used to provide constraints on hydrothermal activity (McInnes et al., 2005). For instance, thermochronometers can be used to discover blind (Hickey et al., 2014) or fossil hydrothermal systems (Person et al., 2008; Luijendijk, 2012) and can be used to quantify the age of hydrothermal systems (Hickey et al., 2014; Gorynski et al., 2014;

Luijendijk, 2012; Márton et al., 2010). While numerous model studies have addressed fluid and heat flow in terrestrial hydrothermal systems on geological timescales (Wieck et al., 1995; Banerjee et al., 2011; Howald et al., 2015; Volpi et al., 2017), the effect of hydrothermal activity on thermochronometer ages has to my knowledge not been modelled. The sole exceptions are Person et al. (2008), who found good agreement between apatite fission track ages around the Carlin gold deposit and ages predicted by numerical models, and Luijendijk (2012), who combined an advective and conductive heat flow model, which was a precursor to the model code presented here, to model heat flow and apatite fission track ages around a hydrothermally active normal fault.

While coupled fluid and heat flow models can provide realistic reconstructions of the thermal history of hydrothermal systems, they are also relatively computationally expensive, which may limit the possibility to explore the response of these systems to different parameters. Heat flow data or thermochronology data are frequently scarce and relatively uncertain. These data can often be explained by a number of different parameter combinations on for instance the age, duration and flow rates of hydrothermal systems. Here I present a new advective-conductive heat flow code, Beo, that can be used to model heat flow and apatite (U-Th)/He thermochronometer ages around hydrothermal systems over geological timescales. In contrast to coupled fluid and heat flow models, fluid flux is prescribed and is therefore not a function of the driving forces of fluid flow and permeability of the subsurface. This makes the code relatively computationally efficient and enables running larger numbers of model experiments with variable parameters. In contrast to inverse thermal models such as HeFTy (Ketcham et al., 2007) and QTQt (Gallagher, 2012) which reconstruct thermal histories of one or more samples, Beo models 2D temperature fields over time, which can be compared to multiple low-temperature thermochronology samples and which allows testing different hypotheses for the history and characteristics of hydrothermal activity.

## 2 Model development

Beo was designed to model advective heat flow in and around a single main fluid conduit. A typical models setup is shown in Fig. 1. The model domain contains a main fluid conduit, which represents permeable fault zone. The main conduit is attached to one or more horizontal fluid conduits that can be used to model lateral flow in and out of permeable formations, that for instance represent alluvial sediments or permeable fractured rocks.

The following sections describe the equations used by Beo to model subsurface and surface heat flux and the apatite (U-Th)/He thermochronometer in hydrothermal systems.

### 2.1 Advective and conductive heat flow

The heat flow equation used by Beo to model conductive and advective heat flow in the subsurface is given by:

$$\rho_b c_b \frac{\partial T}{\partial t} = \nabla K \nabla T - \rho_f c_f \boldsymbol{q} \nabla T \tag{1}$$

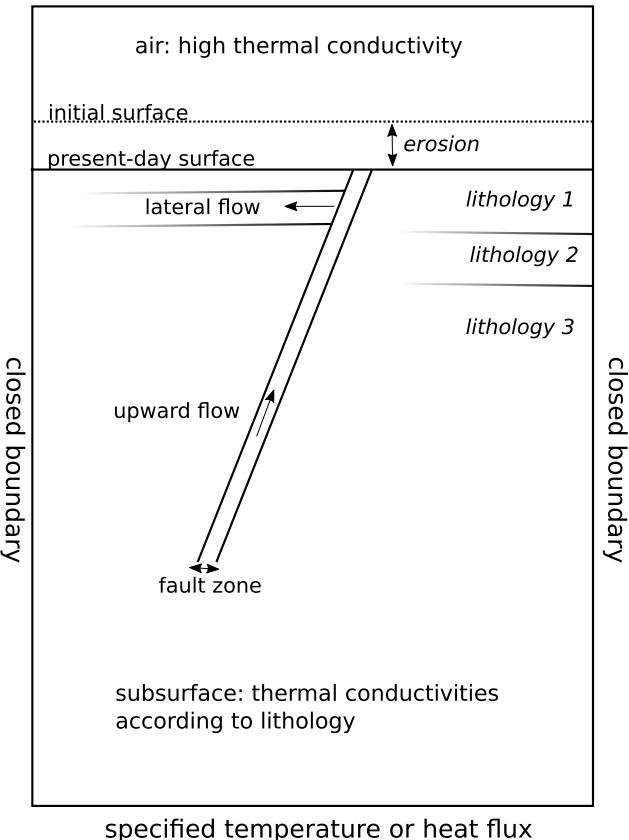

specified temperature

air: high thermal conductivity

initial surface

erosion

present-day surface

lateral flow

lithology 1

lithology 2

lithology 3

closed boundary

closed boundary

upward flow

fault zone

subsurface: thermal conductivities
according to lithology

specified temperature or heat flux

**Figure 1.** Conceptual model setup. The upper and lower boundary conditions were assigned a specified temperature according to the average annual air temperature and the regional geothermal gradient. No heat flow is allowed over the left and right hand model boundary. Fluid flows upwards along a single fault zone, part of the flux contributes to lateral flow in one or more aquifers that are connected to the fault. The remaining fluid discharges at the surface. Heat transfer between the surface and the atmosphere is modelled as a conductive heat flow, with a variable thermal conductivity based on equations for sensible and latent heat flux. Heat transfer in the subsurface is determined by the specific thermal conductivities of the local lithologies. The land surface is lowered over time to account for erosion.

In which $T$ is temperature (K), $t$ is time (s), $c$ is heat capacity (J kg$^{-1}$K$^{-1}$), $\rho$ is density (kg m$^{-3}$), $K$ is thermal conductivity (W m$^{-1}$K$^{-1}$), and $\boldsymbol{q}$ is fluid flux (ms$^{-1}$). Subscripts $_b$ and $_f$ denote properties of the bulk material and the pore fluid, respectively. Beo solves the implicit form of the heat flow equation by discretization of the derivative of $T$ over time:

$$-\nabla \Delta t\, K \nabla T^{t+1} + \Delta t\, \rho_f c_f \boldsymbol{q} \nabla T^{t+1} + \rho_b c_b T^{t+1} = \rho_b c_b T^t \qquad (2)$$

5      where $\Delta t$ is the size of a single timestep (s), $T^t$ (C) denotes the known current temperature and $T^{t+1}$ (C) denotes the temperature at the new timestep t+1. The initial (undisturbed) background temperature is calculated by solving the steady-state

heat flow equation, i.e. the heat flow equation with the term $\frac{\partial T}{\partial t}$ and the heat advection term ($q$) set to zero. Beo uses the generic finite element code Escript (Gross et al., 2007a, b, 2008) to model heat transfer. Escript employs Python bindings to the internal c++ model code and enables the use of multiple processors to increase computational power. Mesh generation was performed using GMSH (Geuzaine and Remacle, 2009) using a Python interface included in Escript. The discretized heat flow equation

was solved using the GMRES solver (Saad and Schultz, 1986).

## 2.2   Land surface heat flux

Typical thermal boundary conditions for subsurface heat and fluid flow models are either a specified temperature or a specified heat flux at the top model boundary, which is usually chosen as the land surface. However, in transient hydrothermal systems it is difficult to simulate realistic temperature using these boundary conditions, especially in cases where fluid discharges at the

surface. Assigning a specified temperature or heat flux would require knowledge of the change in fluid temperatures over time, which is only rarely available.

Temperatures at the land surface are predominantly determined by latent and sensible heat flux (Bateni and Entekhabi, 2012). Beo uses an approach that is to my knowledge new in hydrothermal model codes, and models the heat flux in a layer of air overlying the land surface. The top boundary of the model domain is located several meters in the air, and is assigned a

specified temperature that reflects the average annual air temperature. Latent and sensible heat flux from the land surface are approximated by assigning an artificially high value of thermal conductivity to the layer of air. The thermal conductivity of the air layer is calculated using equations for latent and sensible heat flux described below.

The motivation for including the series of equations below in the model code is to simulate realistic land surface and spring temperatures in transient models. Note that the implementation does not incur high computational demands, the equations are

evaluated for land surface nodes only and in contrast to the heat flow equations in section 2.1 these equations can be solved directly without numerical methods.

Following Bateni and Entekhabi (2012) the sensible heat flux at the land surface is given by:

$$H = \frac{\rho c_a}{r_a}(T_a - T_s) \tag{3}$$

where $H$ is the sensible heat flux (Wm$^{-2}$) $\rho$ is density (kg m$^{-3}$), $c_a$ is the specific heat of air (J kg$^{-1}$K$^{-1}$), $r_a$ is the

aerodynamic resistance (s m$^{-1}$), $T_a$ is the air temperature at a reference level (C) and $T_s$ is the surface temperature (C). This expression can be combined with Fourier's law:

$$q = K_s \frac{\Delta T}{\Delta z} \tag{4}$$

where $q$ is heat flux (Wm$^{-2}$), $K$ is thermal conductivity (Wm$^{-1}$K$^{-1}$), $\Delta T = (T_a - T_s)$ (K). Combining equation 3 and 4 and equalling $H$ and $q$ yields an expression for the effective thermal conductivity ($K_s$) between the surface and the reference level $z$:

$$K_s = \frac{\rho c}{r_a} \Delta z \tag{5}$$

where $\Delta z$ is the difference between the surface and the reference level (m).

Latent heat flux is given by (Bateni and Entekhabi, 2012):

$$LE = \frac{\rho_a L}{r_a}(q_s - q_a) \tag{6}$$

where $LE$ is the latent heat flux (Wm$^{-2}$), $\rho_a$ is the density of air (kg m$^{-3}$), $L$ is the specific latent heat of vaporization (J kg$^{-1}$), which is 334000 Jkg$-1$, $q_s$ is the saturated specific humidity at the surface temperature kg kg$^{-1}$), $q_a$ is the humidity of the air (kg kg$^{-1}$). Combining this with Fourier's law gives the heat transfer coefficient for latent heat flux ($K_l$) as:

$$K_l = \frac{\rho L \Delta z}{r_a} \frac{q_s - q_a}{T_s - T_a} \tag{7}$$

The saturated specific humidity ($q_s$) was calculated as (Monteith, 1981):

$$q_s = 0.622 \frac{e_s}{P_a} \tag{8}$$

where $e_s$ is saturated air vapour pressure (Pa), $P_a$ is surface air pressure (Pa). The saturated air vapour pressure was calculated using the Magnus equation (Alduchov and Eskridge, 1996):

$$e_s = 0.61094 \, e^{\left(\frac{17.625T}{T + 243.04}\right)} \tag{9}$$

Air pressure was assumed to be $1 \times 10^5$ Pa. The thermal conductivity assigned in the air layer is the sum of the heat transfer coefficient for latent heat flux ($K_l$) and sensible heat flux ($K_s$).

The resulting heat flux at the land surface is predominantly a function of the aerodynamic resistance ($r_a$). We use a value of 80 s m$^{-1}$ following values reported for areas covered by short vegetation (Liu et al., 2007), and use a range of ±30 s m$^{-1}$ to quantify the uncertainty of $r_a$ and its effect on modelled spring temperatures.

The calculated heat transfer coefficient shows a strong dependence on surface temperature and aerodynamic resistance, as shown in Fig. 2. This implies that transient numerical models of hydrothermal systems cannot use a fixed heat transfer coefficient at the land surface, because changes in land surface and spring temperature over time change the heat transfer coefficient.

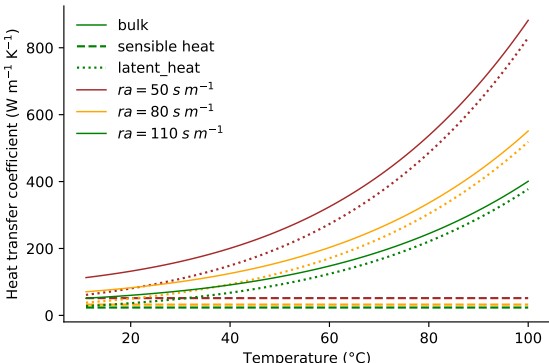

**Figure 2.** Calculated heat transfer coefficient of the air overlying the land surface and its dependence on the temperature of the land surface and aerodynamic resistance (ra). The heat transfer coefficient is an artificially high value of thermal conductivity that takes into account the contribution of latent and sensible heat flux to surface heat flux.

## 2.3  Boiling temperature

The model code simulates conductive and advective heat flow for a single fluid phase. Beo contains an option to cap subsurface temperatures to the boiling temperature curve, which is calculated using a 3rd order polynomial fit to data by the National Institute of Standards and Technology (2018):

$$T_{max} = 3.866 \log(P)^3 + 25.151 \log(P)^2 + 103.28 \log(P) + 179.99 \tag{10}$$

where $T_{max}$ is the maximum (boiling) temperature in the system (°C), and $P$ is fluid pressure (Pa), which is assumed to be hydrostatic. Using the boiling temperature as upper limit ensures that subsurface temperatures remain realistic in a system where vapour is present. Note however that this approach is a simplification in which the latent heat of vaporization is ignored and in which bulk thermal conductivities are not adjusted for the presence of a vapour phase. Therefore, this is intended as a first order approximation of temperatures in multi-phase flow systems, but for more realistic models it would be preferable to use multi-phase flow codes such as Hydrotherm (Hayba and Ingebritsen, 1994). Note that the choice for capping modelled temperature by the boiling temperature avoids the high computational expense of including phase transformations in the model code, which would necessitate solving an additional equation for each node and each timestep, and which would likely place more constraints on spatial discretization and timestep size.

## 2.4  Erosion and sedimentation

For modelling systems that are active over longer timescales Beo can take into account erosion or sedimentation by lowering or raising the land surface over time. This is done in a stepwise fashion, with the default value set to steps of 1 m. The

implementation of erosion is important for low-temperature thermochronometers, because it exposes rocks that have been buried deeper and may have experienced more hydrothermal heating than rocks at the surface that are buffered by surface temperatures.

## 2.5 Apatite (U-Th)/He thermochronology

The modelled temperature history was used to calculate the response in low-temperature thermochronometer apatite (U-Th)/He (AHe). The AHe thermochronometer is based on the sensitivity of the diffusion of helium in apatite minerals to temperature. At high temperatures helium diffuses out at a rate equal or higher than the production by radioactive decay, whereas at low temperatures helium is retained in the apatite minerals. The AHe thermochronometer is sensitive to temperatures ranging from approximately 40 to 70 °C (Reiners et al., 2005). AHe ages were calculated by solving the helium production and diffusion

equation for apatites using the Eigenmode method, following Meesters and Dunai (2002a) and Meesters and Dunai (2002b), which is a computationally efficient method that provides the same results as more computationally demanding finite difference methods (Meesters and Dunai, 2002a, b). Helium production and diffusion is described by:

$$\frac{\partial C}{\partial t} = D\nabla^2 C + S_p U \tag{11}$$

where $C$ is the concentration of helium ($\mathrm{mol\,m^{-3}}$), $D$ is the diffusion coefficient of helium ($\mathrm{m^2 s^{-1}}$)and $U$ is the helium

production rate ($\mathrm{mol\,m^{-3}s^{-1}}$). The term $S_p$ denotes the probability that an emitted alpha particle stops at location x,y,z in the apatite crystal (dimensionless), which is required because the long stopping distance of alpha decay (approximately 21 µm) means that some of the helium that is produced by radioactive decay is not retained inside the crystal. The helium age is calculated using the modelled average concentration of helium in the crystal ($C_{avg}$) following:

$$\text{AHe age} = \frac{C_{avg}}{U} \tag{12}$$

The helium diffusion model assumes a spherical shape of the apatite crystal, and can be compared to measured AHe ages when using an equivalent spherical shape with the same surface to volume ratio as the actual crystal (Meesters and Dunai, 2002a).

The diffusivity ($D$) of helium in apatites depends on temperature and on radiation damage. Radiation damage slows helium diffusion (Shuster et al., 2006; Flowers et al., 2009; Gautheron et al., 2009). The effects of radiation damage on the diffusivity

of helium is calculated using the RDAAM model by Flowers et al. (2009), in which the diffusivity is a function of the density of damage tracks. The annealing of these tracks is dependent on temperature and is assumed to occur at the same rate as fission tracks following established models of fission track annealing (Ketcham et al., 2009, 2007). Beo also includes an option to calculate helium diffusivity following equations by Farley (2000) or Wolf et al. (1998) instead that do not take radiation damage into account.

The parameter $S_p$ is used to correct He concentration in the apatite crystal for the chance that alpha particles that are ejected from locations close to the crystal rim end up outside the crystal. See Meesters and Dunai (2002b) for more details of the

implementation of this parameter in the helium diffusion model. I used an alpha stopping distance of 21 μm (Ketcham et al., 2011).

## 3 Model verification

I first validated the transient conductive heat flow in the numerical model using an analytical solution for the cooling of an intrusive in the subsurface. The solution for temperature change of an initially perturbed temperature field is (Carslaw and Jaeger, 1959):

$$T(z,t) = T_b + \frac{T_i - T_b}{2} \left( \text{erf}\left(\frac{L-x}{2\sqrt{\kappa t}}\right) + \text{erf}\left(\frac{L+x}{2\sqrt{\kappa t}}\right) \right) \tag{13}$$

where $T_b$ is the background temperature (C), $T_i$ is the temperature of the intrusive (C), $L$ is the length of the intrusive (m), $x$ is distance from the intrusive (m) and $\kappa$ is thermal diffusivity ($\text{m}^2\text{s}^{-1}$)

Following Ehlers (2005) I use this equation to simulate cooling of an intrusive in the subsurface as a test case for the model code. The numerical and analytical solutions for cooling are shown in Figure 3. The intrusive body has an initial temperature of 700 °C, and stretches from 0 to 500 m distance. The background temperature is 50 °C. The solutions match to within 1.0 °C.

In addition, the performance of the model code was evaluated using an analytical solution of steady-state heat advection and conduction by Bredehoeft and Papaopulos (1965). The solution describes heat advection in a one-dimensional system with fixed temperatures at the top and bottom boundaries of the model domain. The analytical solution is given by:

$$T = \frac{e^{((\beta z)/L)} - 1}{e^\beta - 1}\Delta T + T_0 \tag{14}$$

and

$$\beta = -c\rho q L/K \tag{15}$$

where T is temperature (K), z is depth (m), $\Delta T$ is the temperature difference between the top and the bottom of the domain (K), $T_0$ is the temperature at the top of the domain (K), c is heat capacity ($\text{J kg}^{-1}\text{K}^{-1}$), $\rho$ is density ($\text{kg m}^{-3}$), L is length of the domain (m) and K is thermal conductivity ($\text{W m}^{-1}\text{K}^{-1}$). A comparison between the numerical solutions by Beo and the analytical solutions shows that the analytical and numerical solutions are identical (Fig. 3 and 4).

## 4 Model usage

### 4.1 Model input

Beo uses a single python file that contains all input parameters. The input parameters are Python variables. The parameter file is organized into two parts, which each part contained in a Python class, named ModelParams and ParameterRanges, respec-

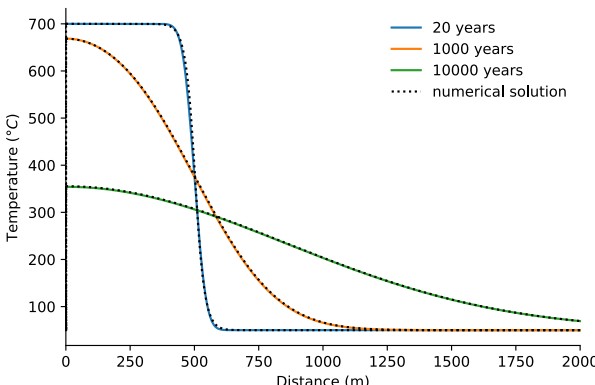

**Figure 3.** Validation of modelled transient conductive heat flow with an analytical solution (Carslaw and Jaeger, 1959) for the cooling of an intrusive over time. Solid lines show calculated temperatures using the analytical solution at three different timesteps. Broken lines show the numerical solution by Beo.

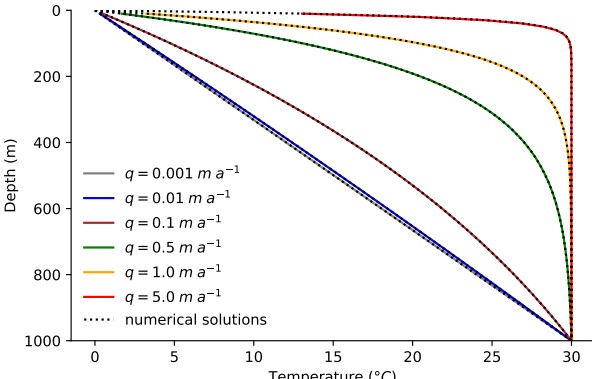

**Figure 4.** Validation of modelled advective heat flow in a one-dimensional system with an analytical solution (Bredehoeft and Papaopulos, 1965). Solid lines show calculated temperatures using the analytical solution and broken lines the numerical solution by Beo.

tively. The class ModelParams contains all the parameters required for a single model run. The parameters cover grid size and discretization, timestep choice, fluxes in a fault and/or connected aquifers, thermal properties of the subsurface and the land surface, and model output options. A manual that contains a detailed description of each parameter and example input data files can be found in the GitHub repository (https://github.com/ElcoLuijendijk/beo/tree/master/manual). The input file can be spec-
5  ified as a command line argument. For instance the command `python beo.py example_input_files/Baden.py` will run the model using the input file `Baden.py` in subdirectory `example_input_files`.

## 4.2 Running multiple models

Beo can be used for automated model runs for model sensitivity analysis or exploration of parameters space. The class ParameterRanges can contain series of parameter values for any parameter from the main parameter set (ie., contained in the class ModelParams). The user can specify to perform model sensitivity analysis, in which each time a single parameter is changed while all other parameters are kept at their base value. Alternatively one can choose to generate model runs for each parameter combination, which can be used to explore parameter space.

## 4.3 Model output and visualization

Beo generates output to comma-separated files, model-specific output files using the Python pickle module, and output of the modelled temperatures and fluid fluxes in VTK format. The comma-separated files contain a copy of all input parameters for each model run, along with several statistics for the model output such as modelled average change in temperatures compared to initial temperatures, modelled temperatures at the surface or user-specified depth slices, modelled AHe data and comparison to observed values. Modelled temperatures and fluxes can be saved as VTK files that can be used for model visualization using external software such as Paraview and Visit. In addition, the model results can be saved in a Beo specific file format that contains all modelled temperature and AHe data. These output files can be used by a separate script (`make_figure.py`) to automatically generate figures such the model results shown in this study (Fig. 5).

## 5 Application

The following section presents models of two active hydrothermal systems. The first case study consists of a model of the thermal history of the Baden and Schinznach spring system, a hydrothermal system and series of hot springs at the boundary of the Jura mountains and the Molasse Basin in Switzerland. This study demonstrates the potential and limitations of the use of spring temperatures and discharge to quantify the depth of fluid conduits, and the use of low-temperature thermochronology to reconstruct the history of hydrothermal activity. The second case study is a model of borehole temperatures and borehole apatite (U-Th)/He data in a hydrothermal system at the Brigerbad spring in the Rhone Valley in the western Alps, which is based on data published by Valla et al. (2016). Note that the aim is not to provide detailed case studies, but to illustrate the possibilities of the model code. In addition to these examples, a separate study that uses the model code to quantify the history of the Beowawe hydrothermal system in the Basin and Range Province has been published as a preprint on EarthArxiv (Louis et al., 2018) and is currently in press in Geology (Louis et al., in press]).

### 5.1 Baden and Schinznach hydrothermal system

### 5.1.1 Model setup

The model of the Baden and Schinznach spring system is based on a model study by Griesser and Rybach (1989). The total heat flux in the Baden and Schinznach spring system that is used as a case study is $2.4 \times 10^6$ W, which was calculated using

spring discharge and temperature data reported by Sonney and Vuataz (2008) and an assumed recharge temperature of 10 °C. With a background heatflow of 0.07 Wm$^2$ the minimum contributing area for the heat output of the springs is $3.5 \times 10^7$ m$^2$. This is slightly above the median value for springs in North America reported by Ferguson and Grasby (2011). The model study presented here therefore represents a terrestrial hydrothermal system with a relatively high, but not unusual heat output.

The area hosts a number of springs with a temperature of 30 to 47 °C (Sonney and Vuataz, 2008), and an average discharge along the fault of $2 \times 10^{-5}$ m$^2$s$^{-1}$. Fluid flow is hosted in a relatively shallow thrust fault that dips around 50 degrees to a detachment level around 1000 m below the surface, which may be connected to a deeper normal fault (Griesser and Rybach, 1989; Malz et al., 2015).

The numerical model is based on a conceptual model shown in Fig. 1. The model only includes the discharge part of the hydrothermal system. Groundwater recharge is much more diffuse than discharge and has a negligible effect on subsurface temperatures in comparison to focused groundwater discharge, as shown for instance by model experiments by Ferguson et al. (2009).

A specified heat flow of 0.07 Wm$^{-2}$ was chosen at the lower boundary (Griesser and Rybach, 1989). For the upper model boundary the air temperature is fixed at 10 °C at an elevation of 2 m above the land surface, and the heat transfer at the land transfer is governed by sensible and latent heat flux following equations 3 to 9. Thermal conductivity was fixed at 2.5 Wm$^{-1}$K$^{-1}$ for the rock matrix, 0.58 Wm$^{-1}$K$^{-1}$ for the pore fluids and the porosity was assumed to be 0.15. The model experiments run for a total of 15000 years. This represents the approximate duration of an interglacial stage. The springs may have been inactive during glacial stages of the Pleistocene. During the last glaciation the Jura mountains and part of the Molasse basin were covered by an ice sheet (Preusser et al., 2011), which may have blocked groundwater recharge and spring flow.

Grid cell size was set to 100 m outside the fault zone, 2.5 m in the fault zone, and 0.5 m at the surface outcrop of the fault, and 10 m in the air layer above the land surface. In total this resulted in 166285 nodes. The models used a timestep size ($\Delta t$) of 50 year. Experiments with smaller timesteps of 0.1 year showed no difference in modelled temperatures.

### 5.1.2 Sensitivity of modelled spring temperatures

A series of model experiments was performed to quantify the effects on the depth, angle and width of fluid conduits on spring temperatures. The base case model assumed a fluid conduit depth of 7 km, a conduit angle of 65 degrees and a width of 10 m. The modelled subsurface and spring temperatures are shown in Fig. 5. The upward fluid flow raises temperatures in a narrow zone around the fluid conduit. After 15000 years the area around the fault zone where temperatures are at least 20 °C higher than the background values is approximately 2000 m wide.

The model results show a strong dependence of modelled spring temperatures on the assumed depth of the fluid conduit (Fig. 6a). The observed temperatures of the Baden and Schinznach spring system are only reached using fluid conduits that are at least 7 km deep. In addition, comparison of a the effects of fluid conduits dipping 50 degrees and 65 degrees show a strong difference in spring temperatures (Fig. 6a). For a fluid conduit with a low dip angle the flow path from a particular depth is longer and therefore the amount of heat loss along the way is higher. Modelled spring temperatures for a fluid conduit of 50 degrees are too low to explain the observed temperatures in the system. The model results confirms the hypothesis proposed

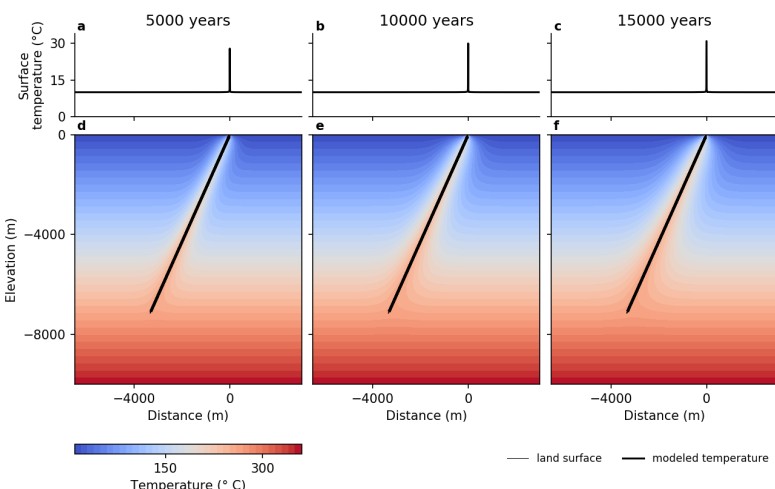

**Figure 5.** Base case model run for the Baden-Schinznach system, showing modelled subsurface (d-f) and spring (a-c) temperatures at three timeslices.

by Griesser and Rybach (1989) that the fluid source in the Baden and Schinznach hydrothermal system is likely a deep and steeply dipping normal fault that is connected to the more shallow thrust fault that hosts the springs near the surface.

In addition to the depth of the fluid conduit spring temperatures are sensitive to the assumed width of the fault zone (Fig. 6b). The wider the fault zone, the lower the spring temperature. Note that in these model runs, the overall flux was kept at $2 \times 10^{-5}$ m$^2$s$^{-1}$, which was redistributed evenly over the width of the fluid conduit. The narrower the fluid conduit, the higher the flow velocity, and the lower the conductive heat loss along the way.

The model experiments also show a strong dependence of spring temperatures on the modelled heat flux at the land surface. The key parameter governing latent and sensible heat flux at the land surface is the aerodynamic resistance (Fig. 2). The value of aerodynamic resistance strongly affects spring temperatures. Lower values of resistance, which correspond to more vegetated conditions (Liu et al., 2007), result in higher values of effective thermal conductivity and heat flux at the surface (Fig. 2), and as a result lead to lower spring temperatures (Fig. 6b).

### 5.1.3   Hydrothermal activity and low-temperature thermochronology

The effect of hydrothermal activity on low-temperature thermochronology was explored by using modelled thermal history of the Baden and Schinznach hydrothermal system to calculate to calculate AHe ages. For models on longer timescales exhumation plays a role. Due to the buffering effect of air temperature rocks at deeper levels heat up much more than rocks close to the land surface (Fig 5). The rate of exhumation therefore determines the strength of the hydrothermal perturbation of thermochronometer ages. The effect of exhumation rates is explored using two different exhumation rates, representing slowly exhuming areas such as passive margins ($1 \times 10^{-4}$ m a$^{-1}$) and moderate exhumation rates representing orogens ($1 \times 10^{-3}$

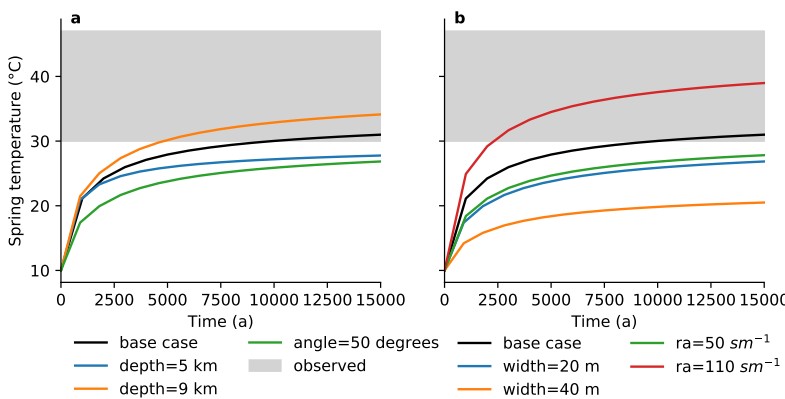

**Figure 6.** Modelled spring temperatures over time and sensitivity of spring temperatures to the geometry of the fluid conduit and the heat flux at the land surface. The observed present-day temperatures in the springs range from 30 to 47 °C (Sonney and Vuataz, 2008). The base case model uses a fluid conduit depth of 7 km, a width of 10 m, an angle of 65 degrees, and an value of aerodynamic resistance (ra), which governs the heat flux at the land surface, of $80 \, \mathrm{s\,m}^{-1}$

$\mathrm{m\,a}^{-1}$) (Herman et al., 2013). Exhumation rates in the northern part of the Molasse Basin where the springs are located is still under debate, but equal approximately 1.5 km over the last 5 million years (Cederbom et al., 2011) or 12 million years (von Hagke et al., 2015) ($1 - 3 \times 10^4 \, \mathrm{m\,a}^{-1}$). Note that to better compare the model results, the initial AHe age, i.e., the AHe age before the start of hydrothermal activity, was set to the same value for both model experiments.

5     The results demonstrate that the effect of hydrothermal activity on thermochronometers is dependent on background exhumation rates. For a model run with a high exhumation rate of $1 \times 10^{-3} \, \mathrm{m\,a}^{-1}$ the width of the zone at the surface where samples are partially reset is 35 m after 15000 years, which is 10 m wider than the width of the fluid conduit (25 m) in this model experiment. This means that low-temperature thermochronometers can be used to quantify the history of active hydrothermal systems and hot springs, but only if sampling is very dense.

10     For samples located at 500 m depth thermochronometers are affected up to 850 m distance from the fluid conduit (Fig. 8). This means that even over a relatively short timescale of one interglacial stage ($\sim$15000 years), hydrothermal activity can affect low-temperature thermochronometers at the subsurface in large areas. The effect of hydrothermal activity may be important for the interpretation of thermochronometers from boreholes near hydrothermal systems or areas close to exhumed hydrothermal systems.

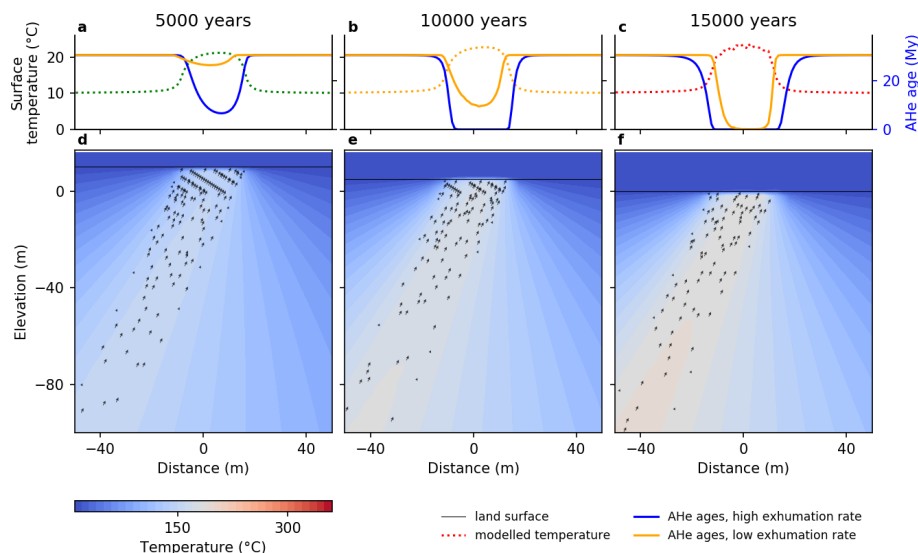

**Figure 7.** Modelled surface and subsurface temperatures and AHe ages for three time slices for a model run with a high exhumation rate $(1 \times 10^{-3} \ m \ a^{-1})$ and a low exhumation rate $(1 \times 10^{-4} \ m \ a^{-1})$. Panels a, b and c show the modelled temperature at the land surface and modelled AHe ages for surface samples. Panels d, e and f show the modelled temperature field in response to upward advective flux along a fault for the model run with a high exhumation rate.

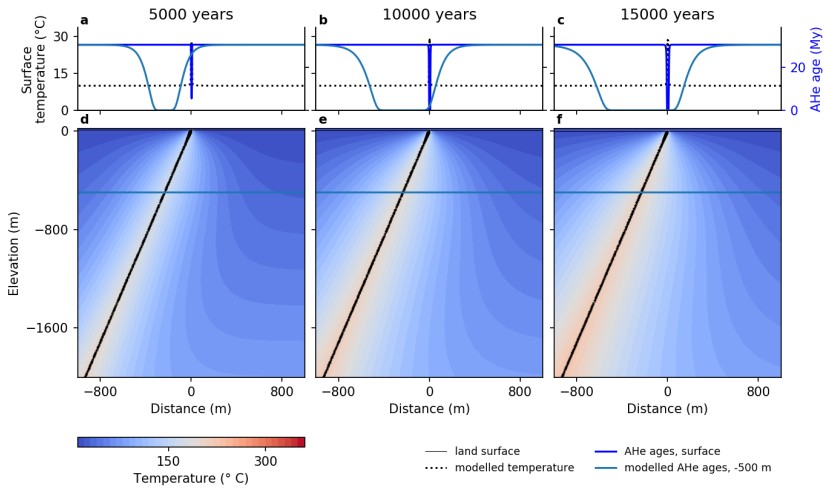

**Figure 8.** Modelled surface (panels a-c) and subsurface temperatures (panels d-f) and AHe ages at the surface and at 500 m depth for three time slices. The model results show the much larger effect of hydrothermal activity on the AHe ages at depth.

## 5.2 Brigerbad

### 5.2.1 Model setup

The Brigerbad hydrothermal system contains three natural springs that are located at the valley floor at the transition from the crystalline bedrock of the Aar massif to low-permeable quaternary valley fill. The springs are fed by a mixture of shallow cold water and a deep groundwater source that is likely channelled upward along a major thrust fault that is located in the valley floor and that forms the contact between the crystalline rocks of the Aar massif and low permeable sedimentary units of the Helvetic nappe (Buser et al., 2013; Valla et al., 2016). Subsurface temperatures in a 500 meter deep borehole close to the springs show a strongly elevated geothermal gradient of approximately $100\,^{\circ}\mathrm{Ckm}^{-1}$ that is caused by the upward flow of warm groundwater.

The hydrothermal system was modelled using a simplified setup that uses a wide fault zone that functions as a flow conduit with a maximum depth of 3.5 km and an upper boundary at a depth of 100 m, where the fault is overlain by low-permeable Quaternary sediments (Buser et al., 2013). The depth of fluid flow is based on an estimated maximum fluid temperature of 110 °C, which is based on geothermometers (Buser et al., 2013). The fault zone is attached to an inclined fluid conduit that channels fluids away laterally from the overlying Quaternary sediments towards the springs that are located at a distance of approximately 125 m north of the fault outcrop. In addition there is a second fluid conduit that channels shallow and cold groundwater from the Aar massif to the north to the same springs and that was estimated to be 100 m deep following general permeability trends in fractured crystalline rocks (Ranjram et al., 2015; Achtziger-Zupančič et al., 2017). The total flux was estimated as $230\,\mathrm{m}^2\mathrm{a}^{-1}$ following reported discharge values of the springs (Sonney and Vuataz, 2008). Following published conceptual models and hydrochemical data (Buser et al., 2013) an estimated 50 % of the total discharge is deep fluid flow through the fault zone and 50 % consists of shallow fluids that flow through the upper 100 m of subsurface from the Aar massif to the north of the springs. The initial, undisturbed geothermal gradient was $27\,^{\circ}\mathrm{C\,km}^{-1}$ following Valla et al. (2016). The thermal conductivity of the crystalline host rock was estimated as $2.5\,\mathrm{W\,m}^{-1}\mathrm{K}^{-1}$ and the porosity as 10 %. The total dimensions of the model domain was 5000 by 5000 m. The mesh contains 116854 nodes, and uses the same cell sizes and timestep size as the model experiments for the Baden and Schinznach hydrothermal system as described in section 5.1. The duration of modelled hydrothermal activity was 10,000 years, which roughly corresponds to the time since deglaciation in the western Alps.

Valla et al. (2016) report three AHe samples from a borehole near the springs at depths of 167, 333 and 497 m below the surface. The samples contained relatively young AHe ages ranging from 0.5 to 0.8 Ma that were interpreted as the result of the background exhumation history and elevated temperatures caused by hydrothermal activity. Background exhumation was implemented following results by Valla et al. (2016) with fast exhumation from 4 to 3 Ma at $1.5\,\mathrm{km\,Ma}^{-1}$, slow exhumation from 3 to 1 Ma at $200\,\mathrm{m\,Ma}^{-1}$ and fast exhumation due to glacial valley carving starting at 1.0 Ma at a rate of $1.25\,\mathrm{km\,Ma}^{-1}$. The temperature history for each sample was calculated by converting exhumation to cooling rates using a geothermal gradient of $27\,^{\circ}\mathrm{C\,km}^{-1}$, adding the initial background temperature based depth of the samples and subsequently combining the initial

temperature history with the modelled temperature history, which covered the last 10,000 years of the thermal history of the samples.

### 5.2.2 Results

Initial model experiments that use a narrow flow conduit that corresponds to the typical width of a fault damage zone for a major fault of 100 m (Bense et al., 2013) fail to provide a good fit the the borehole temperature record and overpredict temperatures at shallow depths (Fig. 9). The model only provides a good fit when deep groundwater is channelled upward along a relatively wide flow conduit, with a size of 500 m providing a good fit to the temperature data (Fig. 10). This confirms models put forward in earlier studies (Buser et al., 2013) that suggest broad distribution of permeable fractures and fluid flow, as opposed to flow that is confined to a narrow fault damage zone. The modelled temperatures do still show a misfit, which suggests that the flow paths may be even more distributed and complex than the simplified model shown here.

The thermal history results in modelled AHe ages that are largely unaffected by hydrothermal activity and that range from 1.4 Ma to 2.1 Ma (Fig. 10d). This is much higher than the measured values in the three samples reported by Valla et al. (2016). The modelled hydrothermal heating during 10,000 years is too short to affect AHe ages. The mismatch indicates that hydrothermal activity was active for much longer than the 10,000 years that were modelled here. The system is likely to have been active periodically, during interglacial stages.

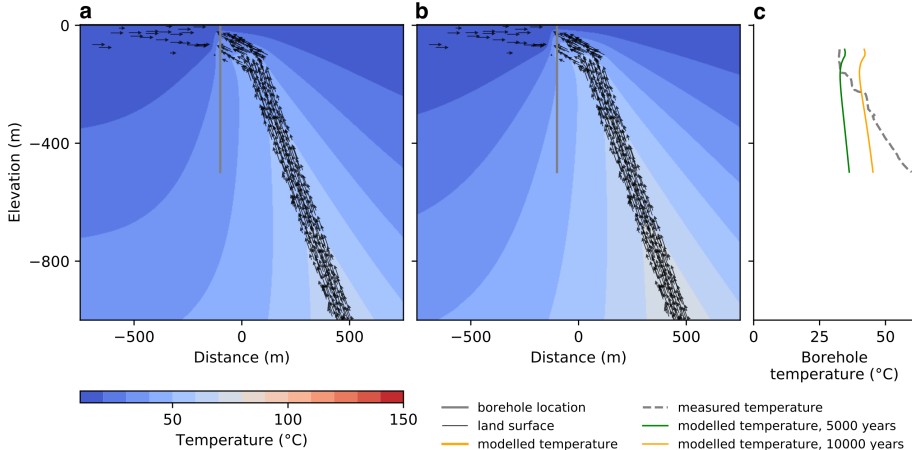

**Figure 9.** Modelled temperatures and AHe ages for the Brigerbad hydrothermal system after 5000 (panel a) and 10000 (panel b) years of hydrothermal activity, assuming a narrow flow conduit of a 100 m. The results show a relatively poor fit to the measured borehole temperature data (panel c).

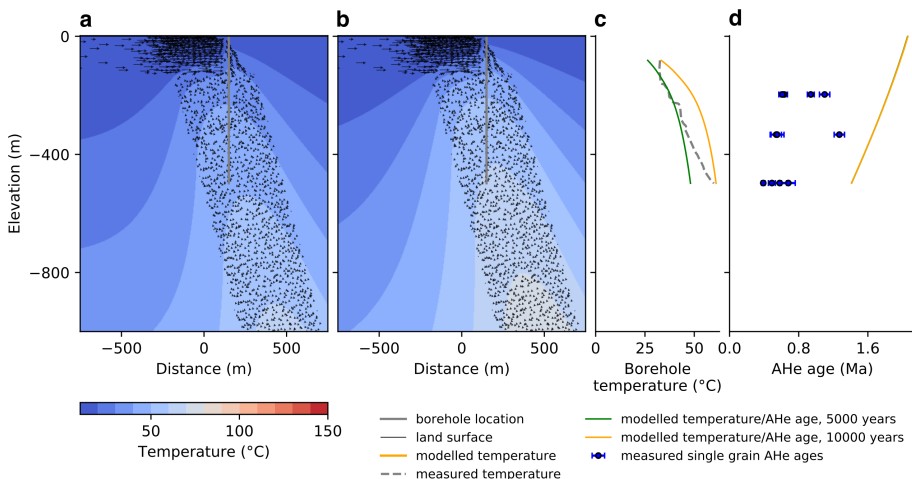

**Figure 10.** Modelled temperatures and AHe ages for the Brigerbad hydrothermal system after 5000 (panel a) and 10000 (panel b) years of hydrothermal activity, assuming a 500-m-wide flow conduit. The results show that a wide flow conduit fits the borehole temperature data much better (panel c), which suggests that a large part of the crystalline bedrock of the Aar massif is permeable enough to channel deep fluids upward. Modelled AHe ages are higher than the observed values (panel d), which shows that one episode of flow is not sufficient to reduce AHe ages.

## 6 Conclusions

Beo v1.0 is an new open-source code for simulating heat flow and apatite (U-Th)/He thermochronology around hydrothermal systems. The model code includes a representation of latent and sensible heat flux at the land surface that provides more realistic spring and land surface temperatures than model codes that use a fixed heat flux, temperature or heat transfer coefficient at the land surface. The code provides new opportunities to quantify the geometry of faults and fluid conduits that are required to match observed discharge rates and temperatures in systems that host thermal springs. The code can also quantify the thermal footprint of hydrothermal systems at the surface and at depth, and provides a tool to quantify the effects of hydrothermal activity on low-temperature thermochronometers. The effects of hydrothermal activity on thermochronometers depends strongly on the duration of activity, and the code provides new opportunities to use thermochronometers to quantify the history of hydrothermal systems and hot springs.

*Code availability.* The source code of Beo version 1.0 has been published at Zenodo (Luijendijk, 2018) and is accessible online (https://zenodo.org/record/2527845). The source code is also available at a GitHub repository (https://github.com/ElcoLuijendijk/beo). The code is distributed under the GNU General Public License, version 3. The repository contains a readme file with a brief description of model installation, usage and output and a more extensive manual that includes a detailed description of all variables in the input data files. In addition, the repository contains several Jupyter notebooks that can be used to reproduce the model benchmarks discussed in section 3,

and the input files for the model examples in section 5. Beo v1.0 depends on the generic finite element code escript (https://launchpad.net/escript-finley) and the Python modules numpy, scipy and pandas.

*Competing interests.* The author declares that no competing interest are present

*Acknowledgements.* The author would like to acknowledge financial support by startup funding for postdocs grant number 3917542 by the University of Göttingen, DFG German Research Foundation grant LU1947/1 and DFG grant LU1947/3, which is part of the priority programme MB-4D. Thanks are also due to Sarah Louis for drafting figure 1 and Sarah Louis, Saskia Köhler and Theis Winter for testing the model code.

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
