# Peer review of "Beo v1.0: Numerical model of heat flow and low-temperature thermochronology in hydrothermal systems."

_Geoscientific Model Development, 2018_

## Referee Comment (RC1) · Peter van der Beek (Referee) · 19 Apr 2019

Luijendijk presents an original modelling approach to assess the advective perturbation of the thermal field along a fault zone that acts as a fluid conduit, and the potential effect on observed surface low-temperature thermochronometer (apatite (U-Th)/He, AHe) ages across the fault zone. The manuscript is concise and clear, the model is original, well documented and open-source, and the potential applications are clear. The author includes two (positive) model-validation tests. I therefore feel that this could make a good contribution to GMD after only modest revision, which could address the following points:

The author appears unaware of the modelling study by Whipp and Ehlers (2007), which, although addressing a slightly different problem (bulk fluid advection through a mountainous rock mass, if such a thing actually exists . . .), was the first to my knowledge to explicitly address the potential effects of fluid flow on thermochronometer ages, and should therefore, I think, be referenced.

Throughout the paper, the reference to "low-temperature thermochronology" is a bit vague – it would be better to state more specifically what you mean, i.e.:

-In the abstract, line 3: ". . . do not include low-temperature thermochronometer age predictions . . ."

-Page 2, line 1: ". . . effect of hydrothermal activity on thermochronometer ages/data . . ."

-Page 2, line 11: ". . . to model heat flow and apatite (U-Th)/He ages . . ."

-etc.

Page 2, line 15: note that the latest version of HeFTy can simultaneously predict thermal histories of multiple (borehole) samples (Ketcham et al., 2018), and that the QTQt code (Gallagher, 2012) also has this capacity.

Page 3, line 2: only two subscripts (b and f), but they refer to three things (the bulk material, the fluid and the solid matrix)?

Section 2.5 on thermochronometer age predictions is relatively condensed. Some more detail could be provided here.

Page 12, line 5: what do you mean by "the strength of the thermochronological signal"? The amount of perturbation? How would you measure this? See also the next comment.

Figure 7 is a key figure as this shows the thermochronometer age predictions. It's a pity that the predicted age pattern is reproduced so small that it is difficult to read.
I wondered how the background AHe age was set in these simulations? It seems strange that the background age is the same (∼30 Ma) for both the high and the low exhumation-rate cases. While this background age seems appropriate for the low exhumation-rate model, the model with high exhumation rate should have a background AHe age that is an order of magnitude smaller (i.e. ∼3 Ma). This is important because the relative perturbation of the age may be similar or even smaller in this case compared to the low exhumation-rate case. This also raises the question of whether such a perturbation could be resolved, either at the surface or within borehole samples. This may need some more consideration.

Linked to this, the model only explores the perturbation during a single interglacial cycle of 15 ky duration. This will only perturb thermochronological systems if very high temperatures (several 100 °C) are reached. Although it has indeed been argued that many hydrothermal systems in Alpine environments would have "switched off" during glacial times, there could still be the cumulative effects of multiple short-lived phases of activity throughout the Quaternary. It might be useful to explore such a scenario.

Linked to the previous two comments; Valla et al. (2016) have reported data from a drill-core close to a hydrothermal site in the Rhone Valley, Switzerland, and have argued that the hydrothermal system was too short-lived to significantly affect the thermochronological ages. They argued that besides limited shortening of fission-track lengths they did not see any effect of hydrothermal circulation on their samples. It could be interesting to use the model presented here to assess this inference more quantitatively.

References:

Gallagher, K.: Transdimensional inverse thermal history modeling for quantitative thermochronology, J. Geophys. Res., 117 (B2), B02408, doi: 10.1029/2011JB008825, 2012.

Ketcham, R. A., Mora, A. and Parra, M.: Deciphering exhumation and burial history

with multi-sample down-well thermochronometric inverse modelling, Basin Res., 30 (Suppl. 1), 48–64, doi: 10.1111/bre.12207, 2018.

Valla, P. G., Rahn, M., Shuster, D. L. and van der Beek, P. A.: Multi-phase late-Neogene exhumation history of the Aar massif, Swiss central Alps, Terra Nova, 28 (6), 383–393, doi: 10.1111/ter.12231, 2016.

Whipp, D. M. and Ehlers, T. A.: Influence of groundwater flow on thermochronometer-derived exhumation rates in the central Nepalese Himalaya, Geology, 35 (9), 851–854, doi: 10.1130/G23788A.1, 2007.
* * *

---

## Short Comment (SC1) · 15 May 2019

My apologies for the late review. I have read the manuscript and since I am not an expert in the field of thermochronology, I will comment on the mathematical and numerical formulation. The paper presents a code solving an advection-diffusion equation for temperature, together with an advection-diffusion equation for helium concentration which is treated as a tracer and complex boundary conditions. The manuscript is overall well written and should be published after addressing minor issues. My main comments are on section 2.

In this section, the author is presenting a complicated combination of constitutive laws

and empirical relationships to provide a better boundary condition for the temperature, accounting for the air layer with its humidity and etc. Despite putting all this detail in the BC however, the author is ignoring phase transformations like vaporisation, which will transform his temperature equation from quasi-linear advection-diffusion to nonlinear advection-diffusion-reaction equation. Rather than that, he is using the boiling temperature curve to cap the temperature. Although he discusses his choice on page 6, line 1-5, the importance of implementing a complicated BC instead of accounting for this mechanism is not obvious to me. I would appreciate if the author could comment on that in the revised version.

In addition, I haven't understood if the code can handle strongly advecting cases, and if yes what kind of unwinding has it been used? All the examples presented seem to be strongly diffusive.

Other than that, I would appreciate if you could explain in more details the transition from Eq. 3 to Eq. 4. I got a bit lost with the nomenclature.

---

## Referee Comment (RC2) · Manolis Veveakis (Referee) · 21 May 2019

My apologies for the late review and the wrong post as a comment. I have read the manuscript and since I am not an expert in the field of thermochronology, I will comment on the mathematical and numer- ical formulation. The paper presents a code solving an advection-diffusion equation for temperature, together with an advection-diffusion equation for helium concentra- tion which is treated as a tracer and complex boundary conditions. The manuscript is overall well written and should be published after addressing minor issues. My main comments are on section 2.

In this section, the author is presenting a complicated combination of constitutive laws

and empirical relationships to provide a better boundary condition for the temperature, accounting for the air layer with its humidity and etc. Despite putting all this detail in the BC however, the author is ignoring phase transformations like vaporisation, which will transform his temperature equation from quasi-linear advection-diffusion to nonlinear advection-diffusion-reaction equation. Rather than that, he is using the boiling temperature curve to cap the temperature. Although he discusses his choice on page 6, line 1-5, the importance of implementing a complicated BC instead of accounting for this mechanism is not obvious to me. I would appreciate if the author could comment on that in the revised version.

In addition, I haven't understood if the code can handle strongly advecting cases, and if yes what kind of unwinding has it been used? All the examples presented seem to be strongly diffusive.

Other than that, I would appreciate if you could explain in more details the transition from Eq. 3 to Eq. 4. I got a bit lost with the nomenclature.

———————————————————

---

## Author Comment (AC1) · 31 Jul 2019

article

**Reply to review by Manolis Veveakis**

Authors' note: The reviewer comments are reproduced here. The replies can be found below each comment and are *italicized*.

My apologies for the late review and the wrong post as a comment. I have read the manuscript and since I am not an expert in the field of thermochronology, I will comment on the mathematical and numerical formulation. The paper presents a code solving an advection-diffusion equation for temperature, together with an advection-diffusion equation for helium concentration which is treated as a tracer and complex boundary conditions. The manuscript is overall well written and should be published after addressing minor issues. My main comments are on section 2.

In this section, the author is presenting a complicated combination of constitutive laws and empirical relationships to provide a better boundary condition for the temperature, accounting for the air layer with its humidity and etc. Despite putting all this detail in the BC however, the author is ignoring phase transformations like vaporisation, which will transform his temperature equation from quasi-linear advection-diffusion to nonlinear advection-diffusion-reaction equation. Rather than that, he is using the boiling temperature curve to cap the temperature. Although he discusses his choice on page 6, line 1-5, the importance of implementing a complicated BC instead of accounting for this mechanism is not obvious to me. I would appreciate if the author could comment on that in the revised version.

*Reply: One of the motivations of writing this model code was to be able to use a heat flow model to quantify what type of hydrothermal system is needed to explain measured discharge temperatures in thermal springs. However, the modelled temperature of a thermal spring is very sensitive to the land surface boundary condition, as discussed in section 2.1 and shown for instance in Figure 6b. Therefore the decision was made to implement a relatively complex but realistic boundary condition that follows standard formulation of land surface heat flux in the meteorology literature. Note that while the boundary condition is relatively complex in the sense that it consists of a large number of equations, in practice the boundary condition places only moderate computational demands. In the code the boundary condition involves calculating a heat transfer co-efficient as a function of the land surface temperature once every modelled timestep for the land surface nodes only. This is a relatively straightforward series of equations*

*that was easily vectorized with numpy. The revised manuscript includes a motivation
for using this boundary condition in section 2.2:*

*"The motivation for including the series of equations below in the model code is to
simulate realistic land surface and spring temperatures in transient models. Note that
the implementation does not place high computational demands, the equations are
evaluated for land surface nodes only and in contrast to the heat flow equations in
section 2.1 these do not require numerical methods."*

*In contrast, including phase transitions would necessitate solving an additional equa-
tion to calculate phase transitions for each node in the model domain. If I am correct
this would also require iterative coupling with the advection-diffusion equation. I have
not come up with a computationally efficient way to include this in the model code, and
suspect that attempt at this will slow down the model code significantly and may also
place additional constraints on grid resolution. Therefore I opted to not invest time in
adding multi-phase flow to the model code for now. This point has been clarified in
section 2.3 by these additional lines:*

*"Note that the choice for capping modelled temperature by the boiling temperature
avoids the high computational expense of including phase transformations in the model
code, which would necessitate solving an additional equation for each node and each
timestep, and which would likely place more constraints on spatial discretization and
timestep size."*

In addition, I haven't understood if the code can handle strongly advecting cases, and
if yes what kind of unwinding has it been used? All the examples presented seem to
be strongly diffusive.

*Reply: The code itself relies on a published finite element code escript for solving the
advection-diffusion equation. The implementation that is used is sensitive to numerical*

*instability that is introduced by advection. To my knowledge escript does not include any unwinding for solving the advection equation, and I did not implement any additional unwinding in the numerical procedure myself. In practice the numerical stability was found to be controlled by timestep size. The solution that was adopted was to start with test models in which the timestep is limited by the CFL condition and subsequently increase the timestep to find the maximum stable timestep size.*

Other than that, I would appreciate if you could explain in more details the transition from Eq. 3 to Eq. 4. I got a bit lost with the nomenclature.

*Reply: I agree that this was not so clear in the previous version of the manuscript. The revised version of the manuscript contains a more extensive discussion and a description of which variables in eq. 3 and 4 were considered equivalent to which variables in Fouriers law.*

---

## Author Comment (AC2) · 2 Aug 2019

article

**Reply to review by Peter van der Beek**

Authors' note: The reviewer comments are reproduced here. The replies can be found below each comment and are *italicized*.

Luijendijk presents an original modelling approach to assess the advective perturbation of the thermal field along a fault zone that acts as a fluid conduit, and the potential

effect on observed surface low-temperature thermochronometer (apatite (U-Th)/He, AHe) ages across the fault zone. The manuscript is concise and clear, the model is original, well documented and open-source, and the potential applications are clear. The author includes two (positive) model-validation tests. I therefore feel that this could make a good contribution to GMD after only modest revision, which could address the following points:

The author appears unaware of the modelling study by Whipp and Ehlers (2007), which, although addressing a slightly different problem (bulk fluid advection through a mountainous rock mass, if such a thing actually exists . . .), was the first to my knowledge to explicitly address the potential effects of fluid flow on thermochronometer ages, and should therefore, I think, be referenced.

*Reply: Thanks for pointing out this reference. I have added a reference to Whipp and Ehlers (2007) to the revised manuscript.*

Throughout the paper, the reference to "low-temperature thermochronology" is a bit vague – it would be better to state more specifically what you mean, i.e.:

-In the abstract, line 3: "... do not include low-temperature thermochronometer age predictions . . ."

-Page 2, line 1: ". . . effect of hydrothermal activity on thermochronometer ages/data ..."

-Page 2, line 11: ". . . to model heat flow and apatite (U-Th)/He ages . . ."

-etc.

*Reply: Agreed. The references to low-temperature thermochronology have been replaced by more specific terms (low-temperature thermochonometer ages) where ap-*

*propriate.*

Page 2, line 15: note that the latest version of HeFTy can simultaneously predict thermal histories of multiple (borehole) samples (Ketcham et al., 2018), and that the QTQt code (Gallagher, 2012) also has this capacity.

*Reply: Agreed, I have adjusted this line to reflect the fact that HeFTy and QtQT can model the history of several samples. The revised version is: "In contrast to inverse thermal models such as HeFTy (Ketcham 2007) and QTQt (Gallagher 2012) which reconstruct thermal histories of one or more samples, Beo models 2D temperature fields over time, which can compared to multiple low-temperature thermochronology samples and which allows testing different hypothesis on hydrothermal activity."*

Page 3, line 2: only two subscripts (b and f), but they refer to three things (the bulk material, the fluid and the solid matrix)?

*Reply: Thanks for noticing this, only two of the subscript were used, this was corrected in the revised version*

Section 2.5 on thermochronometer age predictions is relatively condensed. Some more detail could be provided here.

*Agreed, this section has been expanded in the revised manuscript.*

Page 12, line 5: what do you mean by "the strength of the thermochronological signal"? The amount of perturbation? How would you measure this? See also the next comment.

*Reply: This line was adjusted to: "the strength of the hydrothermal perturbation of thermochronometer ages".*

Figure 7 is a key figure as this shows the thermochronometer age predictions. It's a pity that the predicted age pattern is reproduced so small that it is difficult to read.

*Reply: The figure has been revised to make the AHe age pattern easier to read.*

I wondered how the background AHe age was set in these simulations? It seems strange that the background age is the same (30 Ma) for both the high and the low exhumation-rate cases. While this background age seems appropriate for the low exhumation-rate model, the model with high exhumation rate should have a background AHe age that is an order of magnitude smaller (i.e. 3 Ma). This is important because the relative perturbation of the age may be similar or even smaller in this case compared to the low exhumation-rate case. This also raises the question of whether such a perturbation could be resolved, either at the surface or within borehole samples. This may need some more consideration.

*Reply: For this particular set of models the background AHe age for all the model experiments was the same, regardless of the exhumation rate. This was done to better compare the impact of hydrothermal activity between different model experiments. Visually comparing the size of the zone where AHe ages are partially or fully reset would be more difficult with two different background ages of 30 and 3 My. Note that it would be easier to reset a young AHe age, so i suspect the difference between the two age patterns would be more pronounced if different and admittedly more realistic background ages were used.*

Linked to this, the model only explores the perturbation during a single interglacial cycle of 15 ky duration. This will only perturb thermochronological systems if very high temperatures (several 100 degr. C) are reached. Although it has indeed been argued that many hydrothermal systems in Alpine environments would have "switched off" during glacial times, there could still be the cumulative effects of multiple short-lived phases of activity throughout the Quaternary. It might be useful to explore such a

scenario.

*Reply: While it would be great to explore the effects of repeated hydrothermal activity, i feel that this would deserve a separate study/manuscript in itself. The case studies were meant to demonstrate the abilities of the model code, and adding results on episodic hydrothermal activity would be somewhat unsatisfactory without a thorough discussion, which would detract somewhat from the main objective of the manuscript. In addition, model runs with repeated hydrothermal activity and exhumation tend to require much more computational resources due to the long runtime and the high number of grid nodes required to discretize surface layers and simulate erosion over longer runtime. Such a model run would therefore no longer be in the realm of something that an average desktop computer can run within a reasonable timespan, and is therefore less suited as a demonstration example. Note that we did model the effects of episodic hydrothermal activity for a study in the Basin and Range province that is available on eartharxiv (Louis et al. 2018) and is currently in press in Geology (Louis et al. in press).*

Linked to the previous two comments; Valla et al. (2016) have reported data from a drill-core close to a hydrothermal site in the Rhone Valley, Switzerland, and have argued that the hydrothermal system was too short-lived to significantly affect the thermochronological ages. They argued that besides limited shortening of fission-track lengths they did not see any effect of hydrothermal circulation on their samples. It could be interesting to use the model presented here to assess this inference more quantitatively.

*Reply: Many thanks for the suggestion. The Valla et al study is an excellent demonstration case for the model code and provides an opportunity to compare models with published thermochronology data and also showcases the ability to model borehole temperatures and AHe data. This case study has been added to the revised manuscript, in section 5.2.*

**References**

Louis, S., Luijendijk, E., Dunkl, I., & Person, M. (2018). Episodic fluid flow in an active fault. EarthArXiv. https://doi.org/10.31223/osf.io/cjvxk

Louis, S., Luijendijk, E., Dunkl, I., & Person, M. (in press). Episodic fluid flow in an active fault. Geology.